# Low-Protein Diet: History and Use of Processed Low-Protein Rice for the Treatment of Chronic Kidney Disease

**DOI:** 10.3390/foods10102255

**Published:** 2021-09-23

**Authors:** Shaw Watanabe, Ken’ichi Ohtsubo

**Affiliations:** 1Medical Rice Association, Tokyo 160-0015, Japan; 2Faculty of Applied Life Sciences, Niigata University of Pharmacy and Applied Life Sciences, Niigata 956-0841, Japan; ohtsubok@nupals.ac.jp

**Keywords:** low-protein rice, processed low-protein brown rice, CKD, dietary therapy, lactobacillus

## Abstract

The epidemiology of chronic kidney disease (CKD) shows increasing trends in prevalence and mortality and has become the leading health problem worldwide. Reducing the amount of proteins ingested from rice is an easy way to control the total intake of proteins, saving energy sources, particularly in rice-eating countries. In Japan, low-protein white rice had been developed, but the taste and function were not satisfactory for CKD patients. We reviewed the brief history of low-protein dietary therapy for renal diseases and the recent development of low-protein processed brown rice (LPBR). The new LPBR is characterized by a low-protein content, the same energy content as white rice, low potassium and phosphorus contents, and high amounts of dietary fibers, γ-oryzanol, and antioxidant activity. Dietary fibers and γ-oryzanol would stabilize intestinal microbiota and improve uremic dysbiosis and leaky gut syndrome. All these features suggest that the health benefits of LPBR extend to preventing CKD progression and enhancing the quality of life (QOL) of patients with CKD.

## 1. Introduction

### 1.1. Increasing Trends of CKD in the World

In addition to cardiovascular diseases and cancer, chronic kidney disease (CKD) contributes to increased mortality in many industrialized countries [1,2,3,4]. The prevalence of CKD is estimated to be 8–16% worldwide [5,6,7]. CKD leads to kidney failure, requiring hemodialysis or renal transplantation. Recently, 20–40% of diabetic patients have developed CKD as a late complication [8,9,10]. Adequate interventions are necessary [11,12,13].

In Japan, peritoneal dialysis and hemodialysis are the most current treatments for advanced renal failure, and transplantation is rare. The medical cost is enormous [14,15]: 96% of dialyzed patients receive hemodialysis, and only 3–4% undergo peritoneal dialysis [16,17]. The number of patients treated with hemodialysis in Japan exceeded 344,640 in 2019, including 40,885 newcomers.

To improve patients’ quality of life and decrease medical costs, it is essential to delay the onset of conditions necessary for hemodialysis.

### 1.2. Effects of a Low-Protein Diet on CKD

In 1918, von Noorden W. and Volhard F. reported that a protein-reduced diet (Homburg diet) had suppressed uremic symptoms [18]. Since then, many studies have shown that a low-protein diet (LPD) reduced uremia symptoms, delayed the progression of renal failure, and extended life expectancy [19,20]. In 1963, Giordano reported that a low-protein diet could improve uremic patients’ azotemia and shifted the nitrogen balance from negative to positive [21]. In 1964, Giovannetti also defined a diet whereby animal proteins and energy sources were provided by cereals [19].

In 1983, Brenner reported that excess protein intake imposed a load on glomerular hemodynamics and caused glomerular disorders [22]. After that, the mechanisms of renal damage due to excessive protein intake were rapidly elucidated [23]. Early intervention through LPD could decrease proteinuria in CKD [24,25,26,27,28].

For 50 years, a LPD has been employed to treat chronic renal failure in Japan. It is now recognized that a LPD is almost certainly effective [29,30]. However, large-scale studies conducted by Locatelli et al. did not detect the same effect of a LPD inhibiting renal disease progression [31]. Although low-protein diets have proven very effective for CKD patients, many physicians have failed to control renal dysfunction in RCT; they fail to ensure patients keep to a LPD and maintain their energy intake simultaneously [32,33,34,35]. A recent meta-analysis of RCT showed inconclusive results, and the appropriate protein intake is still an issue of debate [36]. 

Even though excess protein intake promotes glomerular overfiltration and adversely affects renal function, the Japanese guidelines for CKD patients recommend as standard a daily protein intake of 0.8 to 1.0 g/kg for stage G3a and 0.6 to 0.8 g/kg normal weight/day for stage G3b and later [37]. This dose does not seem to be effective, as described below. An extreme LPD was poorly judged to reduce the various risks involved and more work is needed to confirm efficacy and safety in future studies. 

### 1.3. Adequate Amount of Protein Intake by CKD Patients

Historically, the first recommendation on protein intake was 118 g/day/person in the so-called essential diet by German physiologist Professor Voit in 1881 [38]. Professor R.H. Chittenden (Yale University) took a different position. He began to experiment with reduced protein intake in November 1902 on his own body. He first quit breakfast, had only a light lunch, and had a regular supper for seven months until June 1904 [39]. In that time, his intake of protein was progressively reduced. After carefully analyzing nitrogen in his food and urine, Chittenden proved that taking 30–35 g of protein a day was sufficient to maintain his nitrogen balance. This diet also helped Chittenden control rheumatoid arthritis and mild nonspecific complaints. 

Different results had been reported, but from almost all studies, it appeared that simultaneous intake of the right balance of energy and protein seemed to be difficult. In Japan, the dietary reference intake (DRI) uses the reference value of 0.8 g/kg body weight for healthy people. The average protein intake of Japanese people is 0.65 g/kg/day. Adding two standard deviations yields a figure of 0.87 g/kg/day. So, the Japanese DRI was set at 60 g daily for healthy men and 50 g for women [40]. In 1973, the FAO Protein Requirement Committee reported that the minimum physiological input was 0.35 g/kg body weight. So, the lowest acceptable protein intake would be 0.3 to 0.4 g/kg body weight. The experience of Chittenden practicing on his own body and getting better with 30 g a day was substantial evidence in the age of personalized nutrition. After all, Zen priests used to remain healthy with a minimal intake of protein!

### 1.4. Effect of Prolonged Intake of Low-Protein Diet

Many physicians did not adopt LPD therapy, especially after the Evaluation of Modification of Diet in Renal Disease Study (MDRD), in which a LPD resulted in a worse prognosis [41]. Those regimens failed to maintain an adequate energy intake, so malnutrition seemed to worsen the prognosis. 

We carried out a case study to confirm the effect of more than six years of treatment with a LPD. We enrolled ten patients with LPD and hemodialysis patients (0.39 g/kg in CKD and 0.55 g/kg in the hemodialysis group, respectively) or families of the patients and supportive dieticians [42]. The daily protein intake of families was 1.17 g/kg and 1.25 g/kg in the dietician groups, respectively. The recommended protein intake during hemodialysis was 1.2–1.5 g/kg body weight in the guideline, but 0.55 g/kg was enough to maintain body weight and serum protein. Important was the energy intake, being 32 kcal/kg body weight.

Intake of vitamins and minerals was less than half of the DRI, but none of the patients showed signs of deficiency. Study groups did not show any significant difference either in tests using dual-energy X-ray absorptiometry (DEXA), bone mineral density, and non-fatty tissue weight. CKD patients did not complain of sarcopenia, osteoporosis, hyperkalemia, hypo-phosphatemia, or high uric acid. Even lower intakes could reduce the frequency of dialysis in some cases [43].

In another retrospective analysis on 241 CKD patients who participated in the “Low-Protein Diet Practice for advanced CKD” program [44], patients started with serum creatinine levels around 5 mg/dL. Proteinuria improved within a relatively short period, with urinary protein output decreasing by 1.1 g/day after reducing protein intake to 0.5 g/kg body weight. Thus, we recommended LPD protocols, starting with a low-protein diet (0.5 g/kg body weight) from the earliest stage of the disease (eGFR < 60 mL/1.73 m^2^). Blood urea nitrogen (BUN) is a good index for healthy people based on how much they eat protein. A target value was under 15 mg/dL in our population-based cohort study. The substitution of meat with vegetable proteins improved the prognosis in diabetic renal disease [45].

## 2. Low-Protein Diet

### 2.1. How to Design a Low-Protein Diet?

Piccoli et al. investigated a multi-choice system of moderately restricted LPD in more than 4000 human subjects from 2007 until 2015. The three options of LPDs were (1) vegan diets supplemented with alpha-ketoacids and essential amino acids, (2) protein-free bread and pasta, and (3) traditional vegan non-supplemented and tailored. They concluded that a multiple choice of LPD showed good adherence and promising results in dialysis-free follow-up period [46]. Piccoli and Cupisti edited an issue dedicated to low-protein diets in CKD from a global perspective [47]. They told the patients that LPDs are effective and safe, but further efforts are necessary to make them tasty and easy to follow.

To cut down on protein-rich foods, main dishes (meat and fish) should become smaller in size. Generally, a low-protein diet tends to be at the same time a low-energy diet, especially in the Western diet [14]. However, it is difficult to adopt such a degree of frugality during mealtimes. In rice-eating countries, by reducing the 30 g protein content of rice to zero, patients still can eat 30 g protein, which corresponds to 150 g meat, fish, or soy protein. Therefore, making various dishes with meat and fish foods is possible under protein restriction [48].

Protein-adjusted foods started with corn-starch rice, low-protein noodles, and, more recently, processed low-protein white rice. Many brands of so-called ‘low-protein rice’ were actually sold as a retort-type rice made from corn or konjac and processed into the shape of rice. They were not popular among patients. In 2000, Forica Foods Company succeeded in removing proteins from polished white rice using enzymatic treatment. For more than 20 years, this product has been known in Japan as a special therapeutic food with low-protein content [49,50]. 

### 2.2. Structure of Low-Protein Rice 

In the past, when served in hospital meals, low-protein rice was unpopular. It had a strange smell, and patients did not recognize rice’s pleasant taste and texture any more. Nakayama reported that low-protein rice samples with different protein contents had physical properties similar to those of traditional polished rice. Still, it was found to have an unpleasant fragrance upon tasting, so the overall assessment was terrible [51]—there are few studies on these low-protein white rice preparations’ texture, taste, and smell [52].

Untreated rice grains contain a sub-aleurone layer in the periphery and a starchy endosperm toward the center [53]. After protein extraction, the outer layer becomes thin, and starch cells are arranged radially from the center of the endosperm, similar to ordinary rice. Further protein extraction makes the outer layer disappear, and the starch cells inside become indistinct. The process of producing low-protein rice removes the sub-aleurone layer. Consequently, the protein-extracted rice becomes fragile and challenging to cook. 

## 3. Methods to Manufacture PLC

### 3.1. Low-Protein White (Polished) Rice

In 1992, the Niigata Agricultural Research Institute Food Research Center elaborated a technology using *Lactobacillus* fermentation to reduce white rice’s protein, phosphorus, and potassium contents (JP2706888B). However, the method needed technical improvements, due to the long processing time and the difficulty of controlling microorganisms’ contamination. 

In 1994, Forica Foods Co., Ltd. (Uonuma, Japan) developed a proteolysis method utilizing a mixed enzyme solution (JP3156902B) [54]. The following year, the company began to sell cooked PLC Rice 1/3. 

There is no particular restriction on the type of white rice that can be used to make low-protein rice: both *Japonica* rice and *Indica* rice can be used. 

### 3.2. Proteolytic Process

The optimal proteolytic activity requires precise conditions of temperature above 50 °C and acidic pH to prevent uncontrolled microbial proliferation and contamination of the product. The addition of a suitable quantity of citric acid is used to control the pH.

The method employed by Forica Foods was based on an invention by Takahara and Nakajo (JP3156902B), which used proteases to decompose protein. A mixture of enzymes derived from *Aspergillus oryzae, Rhizopus niveus,* and *Aspergillus niger* resulted in better removal and production efficiency. Each enzyme is a crude product extracted from the source microorganisms, and the main component belongs to the aspartic protease family (EC 3.4.23). 

The combination of multiple enzymes, each with different protein cleavage sites, reduces the molecular weight of the processed substrate. 

The protein-extracting process is composed of enzyme digestion and washing out of degradation products (Figure 1). 

### 3.3. Cooking Process

The raw LPR grains are very fragile. Forica Company developed a method for processing LPR into a palatable, high-quality packed rice (WO 2017037799 A1).

After washing out amino acids and short peptides, the rice is dried and boiled under pressured steam in a plastic tray. Steam is injected at high pressure to instantly raise the temperature from 130 °C to 140 °C. Then, hot water is injected to control the hardness of the cooked rice. The steamed LPR is promptly sealed by a heat-resistant film with an oxygen absorber. After cooling, an X-ray inspection checks for foreign matter. The final product consists of individually weighed packages that have passed quality-control tests.

The nutritional aspects of low-protein rice are shown in Table 1. The energy source is preserved, and the protein concentration is controlled well. Low potassium and phosphorus concentrations constitute additional benefits for CKD patients.

The Forica Company registered these low-protein white rice packages with the Ministry of Health, Labour, and Welfare as “Food for Special Use” [49,50] (Figure 2).

## 4. Processed Low-Protein Brown Rice

### 4.1. Technical Challenges with Brown Rice

The beneficial effects of unpolished brown rice have been well recognized [55,56,57,58]. The main benefit of brown rice is the stabilization of the intestinal environment. Significant species found in the microbiota of brown-rice eaters included *Firmicutes* Phila, especially *Blautia wexlerae* and *Faecalibacterium prausnitzi* [59,60]. In chronic renal failure, *Bacteroides* sp. rise, and *Lactobacillus* sp. decrease [61,62], and indole, which is a typical example of uremic toxins, increases. Decreased *Lactobacillus* reduces TLR2 expression on the surface of enteral cells, which leads to loosening tight junctions; decreased tight junctions lead to increased intestinal permeability. This increases the systemic absorption of indole, which is converted to indoxyl sulfate in the liver and exacerbates systemic inflammation. Increased indoxyl sulfate results in elevated cytokines such as IL-6, which are probably involved in cardiovascular complications. Thus, in CKD, two pathological conditions, uremic dysbiosis and leaky gut, result in cardiovascular accidents originating from the intestine [63].

White rice is processed by removing the bran layer of brown rice, which contains most of the functional ingredients, so the idea of manufacturing low-protein brown rice deserved further consideration. Brown rice contains six times more dietary fiber in weight than polished rice and more minerals and vitamins. However, the protein content is high (6–8%). Grains are covered with a strong wax layer, which blocks the penetration of the enzyme solution. In addition, brown rice is usually contaminated with aerobic spores and other germs present in the grain epidermis. 

These represent technical obstacles to the preparation of low-protein brown rice (LPBR) [54,64]. We have recognized that the combination of four steps was essential: rice cultivars, proper grain surface treatment, selection of enzyme solution to extrude rice proteins, and packing method. We were challenged to produce an ideal LPBR by the consortium on the platform in the Ministry of Agriculture, Forestry, and Fisheries. 

### 4.2. Selection of Rice

Rice proteins can be either easily digestible (glutelin and globulin) or indigestible (prolamin) [65]. Albumin and globulin are primarily present in bran, and prolamins are rich in starchy endosperm. Nishimura developed a low-glutelin rice cultivar [66]. 

Mochizuki and Hara used LGC-1 low-protein rice as a staple food (0.6–0.9 g/kg/day) for 23 patients for seven months after a ten-month pre-study period [67]. Nine out of 23 patients consumed rice mainly; protein intake significantly decreased from 47 ± 9 to 42 ± 9 g/day, and the slope of the reciprocal of serum creatinine reduced from −4.59 ± 4.33 to −1.47 ± 3.51 × 10^−4^ mg/dL/day. Mochizuki and Hara concluded that LGC-1 was a valuable and practical food for LPD in CKD patients.

Morita et al. developed PCR markers to detect the *glb1* and *Lgc1* mutations to produce low-protein rice varieties [68]. The *glb1* mutation caused the deficiency of α-globulin, and the *Lgc1* mutation reduced the glutelin content. Combining the *glb1* and the *Lgc1* mutations made it possible to reduce the protein content by approximately 50%. 

Matsui et al. found that final viscosity and consistency were significantly higher in low glutelin lines [69]. The low glutelin lines showed higher amylose content than standard protein composition. These differences were the cause of the low palatability of boiled rice.

So far, low-gluterin rice cultivars, such as the Shunyo and LGC-jun varieties developed in Japan, have had less appealing taste than ordinary rice species. A newly established species of paddy rice, Himeiku-83, cultivated by Ehime Research Institute of Agriculture, Forestry, and Fisheries, was selected for this project. This new variety was a hybrid of Chugoku-188 as mother and Himeiku-71 as the father. Chugoku-188 had both the *Lgc1* gene, which decreased glutelin and *glb1*, which reduced globulin (Figure 3). The protein contents of brown rice range typically between 6.3% and 7.0%. As compared with Shunyo and LGC-jun, Himeiku-83 has a better taste and good grain qualities. The taste of Himeiku-83 is as delicious as Koshihikari, which is the number one brand in Japan. The variety registration was sought for Himeiku-83 on 9 November 2020 [70]. 

### 4.3. Pretreatment and Protein Extraction

The purpose of the polishing process was to reduce the wax layer on the surface of brown rice. It was intended to perform less than 1% refinement at the Mitsuhashi Co. Ltd. (Yoohama, Japan) by making fine scratches on the surface. The surface of brown rice had fine scratches, and the rice seemed to be glassy white.

### 4.4. Protein Extraction

As it is known that heat-stable bacteria (such as *Bacillus* sp.) are present near the aleurone layer of brown rice [71], the epidermis (pericarp and seed coat) was sterilized and washed to eliminate damaged residues in the brown rice. In the sterilization and cleaning process, polished brown rice was placed in 2% aqueous citric acid solution, heated to 70 °C, and stirred to remove impurities, germs, and stains adhering to the surface. Quality control included three cycles of detection of contamination with heat-resistant bacteria in the processed LPBR packs. The final products showed the absence of living bacilli, *Coliform* sp., *Cereus* sp., heat-resistant aerobic bacteria, and anaerobic bacilli [72].

The extraction of proteins included a two-step fermentation (Figure 4). After sterilizing and cleaning the surface with a citrate solution, the sterilized rice was immersed in a solution containing the proteolytic enzyme “Protease M (Amano) G” and *Lactobacillus Plantarum* as primary fermentation agent under anaerobic conditions. The rice was fermented for three days at 40 °C. After draining, a secondary anaerobic fermentation was performed with the same *Lactobacillus*-containing solution for 1.5 days at 40 °C. Then it was washed with water to remove surplus, and after washing with cold water again, it was drained. 

After the fermentation process, a cleaning process was performed, involving draining the secondary fermentation. After that, the rice was washed with running water for 2 h. Fermentation products such as organic acids and proteins generated during fermentation were washed away with decomposing substances and residual lactic acid bacteria. After the removal and washing treatment, the protein content was measured by the Dumas Nitrogen Analyzer NDA701 [73].

### 4.5. Importance of Lactobacillus on LPBR Production

The combination of four steps was the optimal process to produce LPBR. In *Lactobacillus* fermentation, the ability to produce lactic acid is high, and the pH was rapidly reduced in the early stage of fermentation, resulting in an acidic environment. In the optimum pH range, the activity of protease increases, proteolysis is promoted, and a sharply decreased pH suppresses the growth of other bacteria. 

The fermentation tank became anaerobic at an early stage when lactic acid bacteria entered the bran part. Because of carbon dioxide production, a gap that appeared between the bran and the endosperm was quickly filled by enzyme solution and lactic acid bacteria. Lactic acid fermentation and proteolysis treatment at the site were efficient. Further ethanol production suppressed the growth of residual aerobic bacteria that were relatively resistant to lactic acid [74].

### 4.6. Cooking and Packing

For processing into packed rice, trays containing one serving (99 g) of fermented brown rice were steamed for 10 min in a steamer, mixed with 58 mL of hot water (70 °C), and steamed again. After cooking, a seal was applied in the hot state. Each package was sterilized by immersion in a constant temperature water tank at 85 °C for 25 min and then cooled. The pH of the final product of the packed rice was kept at 4.25 by spraying 0.1% gluconic acid.

## 5. Nutrients and Functional Factors of New LPBR

Macronutrient contents are shown in Table 2. Biotech’s LPBR showed a 90% reduction of rice proteins. The proportion of dietary fiber remained two-thirds that of the original brown rice. Water-soluble fibers were more easily resolved than insoluble dietary fibers.

We sent the final rice product immediately frozen to the Nippon Food Analysis Center, Tsukuba, Japan, for nutrient measurement [75]. Energy, major nutrients, vitamins, minerals, γ-oryzanol, and antioxidants (AOU-L and AOU-P) were measured according to the Ministry of Agriculture, Forest, and Fisheries guidelines. SUNTEC Research Laboratories measured the following: vitamin B1, minerals (calcium, phosphorus, iron, sodium, potassium, magnesium, zinc, copper, selenium, and manganate), 28 amino acids, and GABA. Antimony and cadmium were assessed to confirm safety [76].

Oxygen radical absorbance capacity (ORAC) was expressed as Trolox^®^ equivalent [77]. Water-soluble and lipid-soluble AOU (ORAC) was separately measured [78,79]. Relative doses of energy, protein, potassium, phosphorus, dietary fiber, γ-oryzanol, and AOU-h and AOU-l per 100 g packed LPBR are shown in the graphical abstract.

## 6. Nutrient Evaluation of LPBR

LPBR made it easier to adhere to a regimen of 30 g protein/day with enough energy intake. In addition to decreased protein intake, CKD patients need to reduce their phosphorus and potassium intake [32,42]. We employed a simple estimate of the necessary energy, using a simple formula: 0.4× (kg body weight) unit, where one unit corresponds to 80 kcal [80]. LPBR meets all of these requirements as a staple food for CKD patients. Additional benefits of LPBR should be low potassium and phosphorus contents, which prevent hyperkaliemia and hyperphosphatemia [13,58].

A pack of LPBR contained 150 g LPBR (234 kcal), while the protein content was only 0.3 g. When patients ate three packages a day, they could consume 702 kcal and only 0.9 g protein in total. One portion contained 2.25 mg potassium, 76.5 mg phosphorus, and no NaCl. In addition, the dietary fiber contents were 4.5 g, and γ-oryzanol 28.4 mg, which could stabilize the intestinal environment and intestinal bacteria [59,60]. The antioxidant ability is 1350 μmole TEQ; i.e., one-fourth of the daily intake [81]. Many phytochemicals contain antioxidants, which protect from damage caused by free radicals. The U.S. Department of Agriculture has measured the antioxidant capacities of 326 food items. The ORAC values for the hydrophilic fractions (H-ORAC) were much higher than those of the lipophilic fractions (L-ORAC) in fruits and vegetables. L-ORAC remained in LPBR because the wax layer was still present. The consumption of rice as a staple food in daily meals could prevent diseases caused by free radicals [56]. These characteristics make LPBR the ideal staple food for CKD patients. In combination with soy protein for side dishes, it could help to achieve a tasty low-protein diet [82].

## 7. Necessity to Expand the Market

The total annual production (including by Biotech Co., Ltd. (Beijing, China) Forica Foods and Kameda Seika brands) is 18 million packages in Japan. When one patient consumes three packs a day, they will consume 1095 packs a year. This capacity would supply only 18,000 people. There could be several reasons to explain why low-protein rice has not become popular. 

(1) Physicians are not aware of the benefits of LPD. (2) They ignore it deliberately and do not inform patients. (3) A registered dietitian cannot prescribe LPR in Japan. (4) Bad memories about the taste of low protein starch rice have been lasting. (5) Lack of financial market incentive. (6) Information has not reached decision-makers.

We propose some actions to remedy this situation. (1) Wider dissemination of information to the medical community about the benefits of low-protein diets, and planning of as many clinical trials of LPR as we can. (2) Distribution of samples of LPBR to patient groups. (3) Spreading examples of menus based on proper protein diets throughout society. (4) Reliable processing of protein, phosphorus, and potassium, remaining dietary fiber and γ-oryzanol, could be guaranteed by the JAS. (5) Recommending LPBR as medical rice, a low-protein diet that ranks highly. (6) To avoid a shortage of supply, planning of a production system for 100 million packs for 100,000 people. 

The benefits of a low-protein diet could be summarized as follows: preservation of kidney function, improvement of hyperphosphatemia and hyperkalemia, decrease in urinary protein excretion, improvement of subjective symptoms, reduction of complications, and prolongation of the interval between hemodialysis sessions. Control of patients after the induction of hemodialysis is also better by protein restriction [43].

The above effects are expected through the standards for processing low-protein brown rice set by JAS. It is possible to lower the psychological hurdle of a low-protein diet and appeal to the unaffected elderly who suffer from deterioration of renal function with aging. 

## 8. Expansion of the Use of Processed Low-Protein Rice to the World 

CKD has been increasing in prevalence and mortality in many countries and has become a leading health problem. We tried to develop a low-protein Indica rice for clinical trials in Thailand by building a small plant in Kasetsart University in collaboration with Professor Patcharee Tungtrakul in Bangkok. The number of CKD patients increased in Thailand in parallel to obesity, hypertension, and diabetes. We also performed a RCT in Huadong Hospital (Shanghai) with Professor Sun to examine the effect of 12 weeks of dietary therapy for CKD patients [83]. Chinese CKD patients accepted low-protein rice well as a staple food and showed a good response. After 12 weeks consuming a low-protein rice, urine albumin excretion decreased from 130.8 mg/day to 60.8 mg/day (*p* < 0.05).

In both countries, all patients were satisfied with the taste of processed low-protein rice. So, we believe that the processed low-protein brown rice could benefit all CKD patients in rice-eating countries.

Mafra and Leal reviewed a practical approach to the dietetic management of CKD patients in Brazil [84]. As Brazilian people have a high-protein intake, primarily red meat, it is challenging to introduce a low-protein diet (LPD). Rice and beans are also traditional food and essential contributors to the high-protein content of the Brazilian diet, so the reduced protein rice should be helpful. 

Ashuntantang reviewed a practical approach to low-protein diets for patients with CKD in Cameroon [85]. In Cameroon, more than 80% of patients presented late for care, precluding dietary therapies. Implementation of LPBR could become an effective intervention in countries with a weak health system, few dieticians, and limited nutritional expertise.

## 9. Conclusions

The low-protein diet is beneficial for preventing CKD progression. In rice-eating countries, protein-removed rice with low potassium and phosphorus was useful for dietary therapy of CKD patients. Removing the protein from the staple food rice makes it possible to achieve a low-protein diet without changing the composition of side dishes. 

The newly processed low-protein brown rice (LPBR) has the character of a low-protein content, the same energy content as white rice, low potassium and phosphorus contents, and high amounts of dietary fibers, γ-oryzanol, and antioxidant activity. Dietary fiber and γ-oryzanol stabilize intestinal microbiota and improve uremic dysbiosis and leaky gut syndrome. All these features suggest that the health benefits of LPBR extend to preventing CKD progression and enhancement of quality of life of patients with CKD.

The benefits of LPD were preservation of kidney function, improvement of hyperphosphatemia and hyperkalemia, decrease in urinary protein excretion, improvement of subjective symptoms, reduction of complications, and prolongation of the interval between hemodialysis sessions. Control of patients after the induction of hemodialysis was also better by protein restriction. 

## Figures and Tables

**Figure 1 foods-10-02255-f001:**
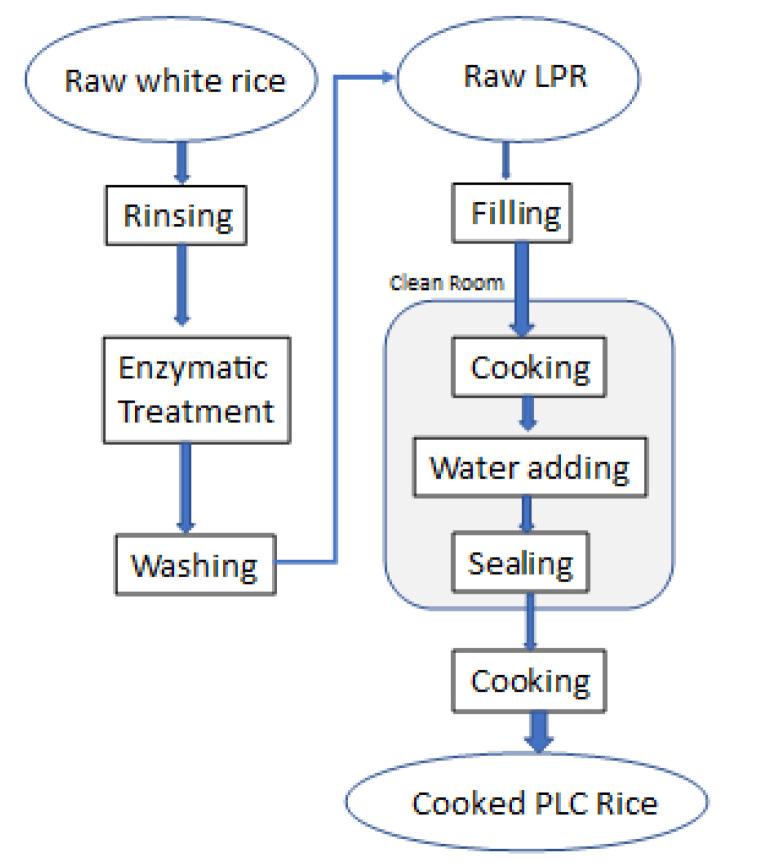
Production flow chart of low-protein white rice. Two-step treatments are necessary: mixing the enzyme solution and cooking and sealing. In the first step, surface bran residues and germs are washed away with water. The washed white rice is placed in a reaction solution containing dissolved enzyme mix and citric acid and allowed to soak for a certain period (up to 24 h) at a temperature > 50 °C and pH < 3.0 conducive to enzyme activity.

**Figure 2 foods-10-02255-f002:**
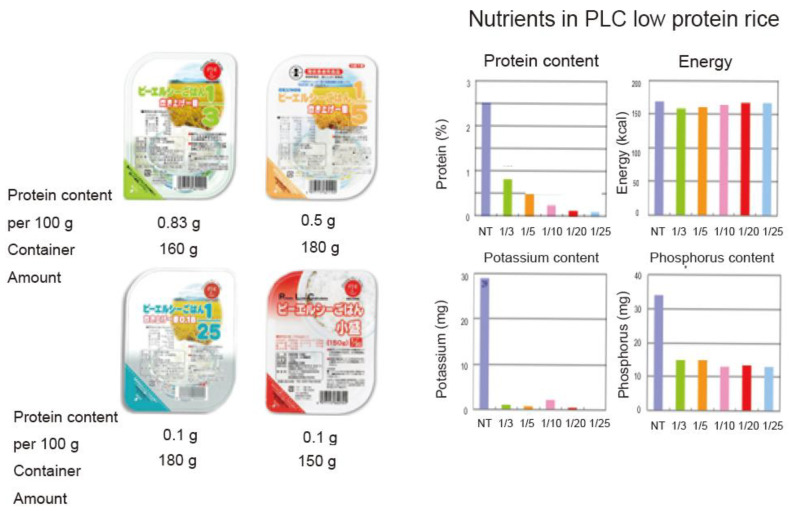
Various PLC rice products. The label claims that they are suitable foods for kidney disease patients. When launched in 1995, the first product contained 1/3 of the protein in WR. With continuing technological developments, the company achieved a reduction rate of 1/25 by 2007. As of 2020, the PLC rice series expanded to include ten products with different degrees of protein reduction, from one-third to one-25th, and two serving sizes (140 and 180 g).

**Figure 3 foods-10-02255-f003:**
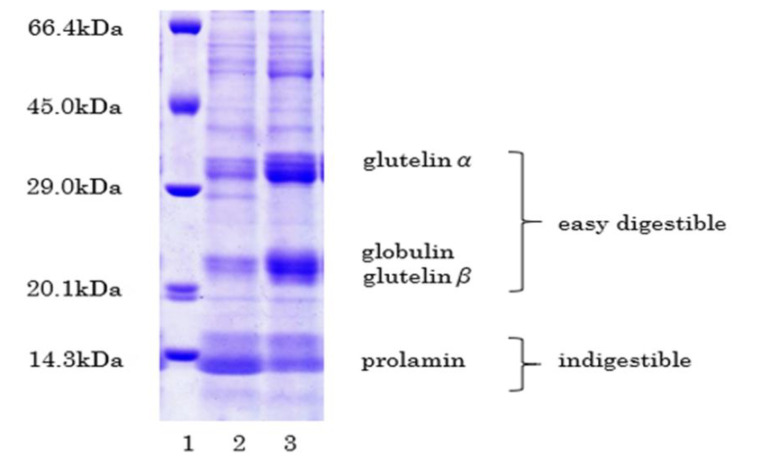
Western blot of Himeiku-83. 1: molecular maker; 2: Himeiku-83; 3: Hinohikari.

**Figure 4 foods-10-02255-f004:**
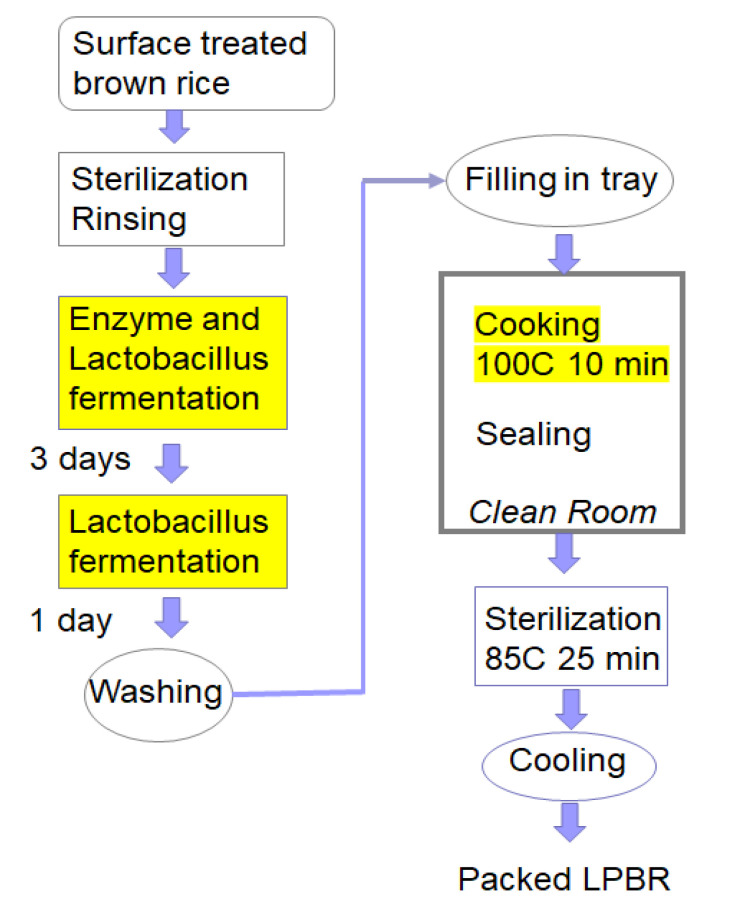
Steps to produce a processed low-protein brown rice.

**Table 1 foods-10-02255-t001:** Variety of PLC products from Formica Co. Ltd. (Uonuma, Japan).

	White Rice	Protein Extracted Rice									
Content/100 g Cooked Rice	Origin	1/3		1/5		1/10		1/20		1/25	
		mean	sd	mean	sd	mean	sd	mean	sd	mean	sd
Energy (kcal)	168	160	3.1	162	6.8	162	2.6	160	7.1	160	1.7
Moisture (%)	60.0	60.4	0.7	60.1	1.7	60.1	0.8	60.5	1.6	60.6	0.5
Protein (%)	2.5	0.87	0.1	0.44	0	0.24	0.1	0.13	0	0.1	0
Fat (%)	0.3	0.5	0.1	0.5	0.1	0.6	0.1	0.3	0.1	0.5	0.2
Carbohydrate (%)	37.1	38.1	0.7	38.9	1.7	39.0	0.7	39.1	1.4	38.8	0.7
Ash (%)	0.1	0.1	0	0.1	0.1	0.1	0.1	0	0	0	0.1
Sodium (mg)	1	2	0.3	2	0	2	0.2	2	0.1	2	0.1
Potassium (mg)	29	0.6	0.3	0.5	0.2	0.9	0.2	0.6	0.1	0.7	0.2
Calcium (mg)	3	5	0	5	0.3	5	1.2	5	0.7	5	0.6
Phosphorus (mg)	34	15	1.0	15	1.5	14	1.0	13	0.7	14	1.2

**Table 2 foods-10-02255-t002:** Nutrients in the new LPBR.

Nutrients/100 g Cooked Rice	Brown Rice *	LPBR by Method	
		Enzyme	Lactobacillus
Energy (kcal)	244	142	156
Water (g)	40.7	65.5	62.2
Protein (g)	1.3	0.5	0.2
Lipid (g)	1.9	1.1	1.3
Ash (g)	0.1	0.1	0.1
Carbohydrate (g)	57.1	32.9	36.3
Sugar (g)	55.6	32.1	35.3
Dietary fiber (g)	1.5	0.8	1.0
g-oryzanol (mg)	10.4	7.9	6.3
Sodium (mg)	2.5	1.6	1.2
NaCleq (g)	0.0041	0.0041	0.003
K (mg)	85.3	nd	0.5
P (mg)	115	nd	14.8
Ca (mg)	6	nd	6
Mg (mg)	47.5	nd	2.2
Mn (mg)	0.83	nd	0.05
Zn (mg)	0.76	nd	0.12
Fe (mg)	0.4	nd	<0.1
Cu (mg)	0.1	nd	<0.1

* means /100 g boiled rice.

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
