# Peer review of "Low-Protein Diet: History and Use of Processed Low-Protein Rice for the Treatment of Chronic Kidney Disease"

_foods, 2021, doi:10.3390/foods10102255_

Round 1

Reviewer 1 Report

In this study, the authors comprehensively reviewed current evidence about the benefit of low protein diet and low protein rice in retarding CKD progression and delaying the requirement of dialysis. The authors introduced a relatively new product- low protein brown rice and declared that the low protein brown rice is even better than currently popular low protein white rice in its higher dietary fiber and anti-oxidative effect. The reviewer recognized this is an important issue to raise general attention on the effect of low protein rice in delaying CKD progression. However, some concept of the association between low protein diet and CKD treatment mentioned in this manuscript is not quite clear. Thus, the reviewer raised some concerns to improve the quality of this review. 

  1. Because low protein diet is not a proved way to “prevent” CKD and it is rarely used this kind of treatment in general population to “prevent” CKD. The reviewer suggested deleting the word “prevention” in the title.
  2. In abstract, the author mentioned that reducing the protein from rice can control both protein and energy sources. Because to keep adequate energy intake while using low protein diet is a crucial part of dietary therapy and we should maintain energy intake instead of controlling it. The sentence here may mislead readers. Thus, the reviewer suggested deleting the word “energy” (line:10)
  3. In line: 57 “It is now recognized that low-protein diets were almost certain” This is an unclear sentence. Do you mean the effect of low-protein diet were almost certain?
  1. From line 97 to120 (including table 1), the authors introduced their own cross-section study trying to elucidate the safety and beneficial effect of low protein diet. However, the design of this study is quite strange to compare the effect of different dietary protein across different populations. In addition, the cross-section design made it impossible to compare the hard outcomes, such as risks of mortality and infection, which are crucial for CKD and ESRD populations. Since this research has limited effect in proving the benefit of low-protein diet, the reviewer suggested deleting the whole paragraph and the table 1.      
  2. In line: 125, the author suggested the low protein diet < 0.5gm/kg/day should be initiated as soon as eGFR< 60ml/min/1.73m2 and BUN is an ideal monitor tool and is needed to keep below 15. These concepts are not generally accepted among nephrologists or nutrionists. Please attenuate these sentences and remove the BUN part (using BUN as index of dietary treatment is not reasonable, BUN will increase along with CKD progression).
  3. The Figure 5 is quite strange, the authors put all items with different units in one figure and thus lead to some items, such as protein, is hard to read. How about taking brown rice as reference (100%) in each item? For example: in energy brown rice 244kcal (reference) LPBR 156kcal (63%).
  4. In conclusion line 480, the authors mentioned “ to increase the number of dialysis patients”, do they meant “to decrease”? In addition, actually, low protein diet can’t really decrease the numbers of CKD patients, but it can retard the progression and delay the initiation of dialysis. Please rephrase the sentence (line 480-482)
  5. Since the authors declared the low protein brown rice is better than low protein white rice, which is currently common-used product. The authors may emphasize this point in conclusion.

Author Response

Thank you for your review.
The modification has been completed.
Please refer to the attachment for the reply.

Reviewer 2 Report

Watanabe and Ohtsubo report an interesting update on the recent techniques used to produce low-protein brown rice (LPBR) illustrating its relative benefits as a part of low-protein diet regimen for patients suffering from chronic kidney disease (CKD).

General comments:

English language is sufficient although needs general revisions. Throughout the manuscripts several repetitions are present: line 15-18 ; line 111-114; line 281-283. The contents are detailed but sometimes are confused (for example regarding the real effectiveness of the low protein diet for CKD). References could be replaced with more recent ones.

Abstract

The abstract is missing the part in which the authors illustrate the type of review and briefly summarize the structure of the article; it may be useful for the reader to have a clearer idea of the content that is exposed in the manuscript.

Main text

Introduction

Line 26-30: epidemiological data regarding CKD are relatively old and referred to the Japanese population mostly, it might be useful to cite more up-to-date and complete data.

Line 62-63: “A recent meta-analysis of RCT showed inconclusive results, and the appropriate  dose of protein intake is still a matter of debate” reference is missing .

Paragraphs 7 and 8 appear a bit pretentious and too biased towards the LPBR as the only resource in the context of low protein diets.

The conclusions are not very structured, please re-elaborated

Author Response

(The authors gave the same response as above.)

Round 2

Reviewer 1 Report

I have no more questions for the authors.